# Total Recall: Lateral Habenula and Psychedelics in the Study of Depression and Comorbid Brain Disorders

**DOI:** 10.3390/ijms21186525

**Published:** 2020-09-07

**Authors:** Matas Vitkauskas, Ajay S. Mathuru

**Affiliations:** 1Yale-NUS College, Singapore 637551, Singapore; matas.vitkauskas@u.yale-nus.edu.sg; 2Institute of Molecular and Cell Biology (IMCB), Singapore 637551, Singapore; 3Department of Physiology, Yong Loo Lin School of Medicine, NUS, Singapore 637551, Singapore

**Keywords:** lateral habenula, psilocybin, depression, comorbid brain disorders, cerebral organoids

## Abstract

Depression impacts the lives and daily activities of millions globally. Research into the neurobiology of lateral habenula circuitry and the use of psychedelics for treating depressive states has emerged in the last decade as new directions to devise interventional strategies and therapies. Several clinical trials using deep brain stimulation of the habenula, or using ketamine, and psychedelics that target the serotonergic system such as psilocybin are also underway. The promising early results in these fields require cautious optimism as further evidence from experiments conducted in animal systems in ecologically relevant settings, and a larger number of human studies with improved spatiotemporal neuroimaging, accumulates. Designing optimal methods of intervention will also be aided by an improvement in our understanding of the common genetic and molecular factors underlying disorders comorbid with depression, as well as the characterization of psychedelic-induced changes at a molecular level. Advances in the use of cerebral organoids offers a new approach for rapid progress towards these goals. Here, we review developments in these fast-moving areas of research and discuss potential future directions.

## 1. Introduction

Depression affects millions of people worldwide. Analysis of the data collated by the World Health Organisation suggests that depression as a leading cause of disability has grown worldwide and in 2017 had more than 250 million sufferers, making it one of the most common mental ailments [1]. Depression has often been observed to be comorbid with other neuropsychiatric ailments and brain disorders. Epidemiological evidence suggests a co-occurrence with nicotine dependence [2,3], substance use disorders [3,4], anxiety disorders [5], insomnia, and chronic pain [6]. Further, patients with comorbid disorders have comparatively worse therapeutic outcomes in treatments [5,7,8]. The neurobiological, genetic, and circuit-level commonalities between these conditions are anticipated but are not well understood at present [6]. Unraveling these commonalities in molecular mechanisms and brain circuits is thus likely to give us an insight into both depression as a brain disorder as well as into the potential therapeutic routes to take for intervention.

In recent times, there has also been an increasing appreciation for the dynamic nature of neural circuitry under duress and the potential role of neuroplasticity that becomes maladaptive in conditions like major depressive disorder (MDD) [9]. This interpretation associates the clinical symptoms of depression with dysfunctional molecular and cellular changes in critical neural circuits in the brain that regulate emotion. One region of the brain that has received an unusual amount of interest in the past few years in this context is the habenula complex [10,11,12,13,14,15]. The habenula complex has been variously described as a “conductor hidden in the orchestra” [16], and as a structure at the “crossroads between basal ganglia and the limbic system”, vital for decision making in vertebrates [17]. One reason to describe the habenula using these metaphors is the regulatory control it exerts in the release of several neuromodulators—serotonin, dopamine, noradrenaline, and histamine among others [18,19]. The habenula complex has two subnuclei, which are biochemically, histologically, transcriptionally, and functionally distinct. Both the lateral and medial habenulae malfunctions have been implicated in several neuropsychiatric disorders. Among the two, the lateral habenula has primarily been associated with depression, while the medial with substance use disorders [11,20,21]. There is growing enthusiasm for the possibility that the lateral habenula is a key player in MDD based on a large number of studies in rodent models of depression [11,14,22,23,24,25,26]. Alongside this body of research, the lateral habenula has become the anatomical target for testing antidepressants such as ketamine infusions in animals [13,14] and its action in humans [27]. Clinical studies of Deep Brain Stimulation (DBS) targeting the habenula in humans [28,29,30,31,32,33] are also underway based on favorable results in rodents [34,35,36]. As promising as this direction of research is, it is still early days and there is a need for cautious optimism as not every observation is fully explained by current theory about how habenula dysfunction could result in depressive states [10]. 

Interestingly, a very different approach from that of targeting a specific brain region has reemerged in parallel in the search for an appropriate antidepressant treatment regimen [37]. This renaissance in the use of psychedelics in psychiatry for pharmaco-resistant depression has come to prominence because of dramatically different treatment procedures and effects documented compared to other pharmacological agents [38,39,40,41]. In one account, a single dose of psilocybin appears to have lowered the severity of an MDD with effects lasting for 5-weeks or longer [38]. The antidepressant effects of psilocybin are proposed to be brought about by a “reset” of the default mode network of the brain in this study [38]. While how such a long-term change in circuits spanning the entire brain occurs at a physiological and molecular level is not fully understood [37,42], clues exist about its action via the serotonergic system. It is interesting to note that psilocybin has also shown promise in the treatment of addiction [43,44]. The co-occurrence of the two disorders and the potential for a common treatment that is effective for both is suggestive of shared underlying affected mechanisms [3]. 

In this article, we review recent developments in these two fast-moving areas of research and discuss the potential future directions that can bridge progress in both. 

## 2. Lateral Habenula, “Negative-Reward”, and the Relation to Depression

The revival of interest in the habenula complex in the context of neuropsychiatric ailments has been fueled by the seminal work conducted by Matsumoto, Hikosaka, and colleagues that demonstrated the reward-prediction error function of lateral habenula [45]. That is, the lateral habenula neurons fire when expected rewards are omitted, and are silenced with unexpected rewards. Their results [46,47,48,49,50] in a behaving monkey have helped contextualize older observations of indirect modulation of the substantia nigra and ventral tegmental areas by the habenula, primarily resulting in the inhibition of dopaminergic neurons [51,52], and to the serotonergic dorsal raphe nucleus, resulting in the regulation of serotonin release [15]. As these immediately downstream regions play a critical role in reward processing and reinforcement learning, these discoveries, in addition to the discovery of functional conservation of this complex in the vertebrate lineage, have sparked a resurgence in the interest in the habenula complex [18,53,54,55]. 

Malfunction in the reward processing and reinforcement behaviors are closely tied to many brain disorders including addiction and MDD [25]. Consequently, animal models where the neurobiology of such behaviors can be studied have also emerged as powerful tools for the study of depression [11,34,55,56,57,58]. A recent study, for instance, examined passive coping behavior in zebrafish while applying brain-wide imaging techniques that allow high-speed imaging of the entire brain at the resolution of single neurons [59]. Passive coping (or learned helplessness) occurs when animals are exposed to a prolonged behavioral challenge, in this case, an inescapable shock regiment. Such coping has been equated with despair in human patients, a common feature of MDD. Analysis of the whole-brain Ca^2+^ responses of neurons revealed that a passive coping strategy was associated with progressive activation of neurons in the ventral habenula (equivalent to the lateral habenula in humans). Conversely, activation of ventral-habenula neurons using optogenetic manipulation, resulted in both behavioral passivity and suppression of the serotonergic neurons in the raphe [59], thus connecting the dots between the neurophysiological response of the habenula neurons and the behavioral outcome. Congenitally learned helpless (cLH) rats and mice, where a similar rationale was applied, have also been used to examine the response of the habenula under conditions of normalcy and coping [14]. A whole-cell patch-clamp in this study showed that 23% percent of lateral habenula neurons in cLH rats display a bursting type activity while the percentage of such firing was only 7% in controls. Moreover, the number of spikes in each burst was also higher at 43% to 7% in controls [14]. A series of experiments from several other groups of researchers has added vastly to our knowledge of the habenula neurophysiology in this context in the last few years [58,60,61,62]. Together, these studies suggest that a hyper or overactivation of the lateral habenula is involved in the animal phenotypes associated with depression. 

## 3. Targeting the Hyperactivity of Lateral Habenula Provides Antidepressant Effects in Animals

Preclinical animal models provide a valuable avenue to dissect a complex phenomenon such as depression and discover potential antidepressant treatments. In rodents, several phenotypes in specific behavioral tests have been used to model human depression, though not all of them may have a face, predictive, and construct validity, and the interpretation of the animal’s behavior in these tests has changed over time [63]. Among them, time of immobility in forced swim tests, sucrose consumption, exploration in an open field, and participation in reinforcement behavior associated with rewards, alongside measurement of neuromodulator concentration, have been used as readouts to quantify coping abilities, the effectiveness of antidepressants, and depressive-like behaviors. One therapeutic angle being tested using these readouts is to find the exact anatomical location for implanting electrodes that deliver deep brain stimulation (DBS) [36]. High-frequency stimuli ranging between 100 and 200 Hz applied either intermittently, chronically over days, or acutely up to an hour before the behavioral tests are conducted have been found to be effective in alleviating readouts that are considered depressive in these animals [28]. Studies show that DBS applied to the lateral habenula of rats over 28 days improved the exploration of open field and sucrose consumption in chronically stressed rats [34]. A study examining maternal separation-induced depressive-like behaviors in rodents also found that a high-frequency DBS to the lateral habenula attenuates the behavioral symptoms [60]. The authors, in this case, also demonstrated that the DBS procedure reduces the lateral habenula hyperactivity [60]. Further, in support of the hypothesis that high-frequency DBS likely works by reducing the hyperactivity, one study showed that low-frequency stimulation conversely activates lateral habenula and produces an increase in depressive-like behaviors in rats [35]. The exact mechanism of DBS action is still not clearly understood, but could occur either because of a change in the presynaptic neurotransmitter release mechanisms [22,24,34] or due to the downstream effects [52]. The behavioral symptom relief in most studies is correlated with an elevation in serotonin, dopamine, and norepinephrine levels [34]. 

Another approach to target the lateral habenula hyperactivity has been the use of pharmacological aids. The two most notable examples are the administration of protein-phosphotase 2A (PP2A) inhibitors and ketamine. Mice experiencing inescapable shock develop depressive-like phenotypes and exhibit neuronal hyperactivity in the lateral habenula [61]. The authors found that one reason for the hyperactivity was a reduction in the surface expression of a G protein—gated inwardly rectifying potassium channels and GABA_B_ receptors, regulated by the enzyme PP2A. An increase in the PP2A activity due to the shock paradigm increases the internalization of these proteins. Inhibition of PP2A, specifically in the lateral habenula, on the other hand, restores the currents from these channels and reverses both the effects on neuronal excitability, and the depressive-like behaviors exhibited [61]. 

A pair of studies from the Hailan Hu lab has also charted out another physiological dysfunction in the lateral habenula and how ketamine rapidly reverses this deficit in rodent models to produce antidepressant effects [12,14]. Ketamine is known to be an NMDA receptor antagonist and has been reported to produce rapid antidepressant effects even when applied systemically [27]. The authors reasoned that ketamine could be acting at the lateral habenula given that the brain region receives glutamatergic afferents and controls neuromodulators implicated in depression [64]. A direct infusion of ketamine converted the bursting activity seen in a subset of lateral habenula neurons of cLH mice to the tonic firing more common in control animals [14]. The group further discovered that the bursting firing pattern in the habenula neurons was dependent on a change in the resting membrane potential of these neurons, which in turn was occurring due to a change in the expression levels of the channel Kir4.1 in the astrocytes that wrap these neurons [12]. Another recent study also suggests the same site of action for ketamine but attributes the effects to the function of mu-opioid receptors (MOR) and NMDA receptors (NMDAR), both of which are highly expressed in the lateral habenula [13]. Ketamine could be acting on many other neurons in the brain with NMDAR and MOR expression, but these observations together suggest a potential explanation for the rapid antidepressant effects of ketamine in comparison to alternatives like SSRIs since the site of action is upstream and can directly correct the putative monoamine imbalance linked with depressive states. 

Targeting the hyperactivity of the lateral habenula, either through DBS or chemogenetic inhibitors [60], or by using drugs [11,36], has been highly effective in animals in reducing depressive-like behaviors, but the suitability of applying these directly to humans raises both ethical and technological considerations. Only a handful of clinical trials of DBS with a small number of subjects have been initiated in humans until now [33,65]. The initial findings are promising, but further studies with larger numbers of subjects will be needed to understand how individual variability impacts humans, before a more complete picture can be constructed. New findings such as the possibility of using non-invasive light therapy delivered through the retina to modulate lateral habenula activity open additional avenues to combine with existing approaches for clinical applications in humans that need further exploration [66]. 

## 4. Lateral Habenula Activity in Human Mdd Patients

Human habenula studies have generally been in agreement with the findings in animals and non-human primates. Habenula activation was reported when there was a punishment prediction error, and in aversive events rather than in neutral or rewarding events [67,68,69]. Functional studies of the lateral habenula in humans, however, are difficult to perform in general in humans as anatomically, both medial and lateral habenula together occupy a volume of approximately 30 mm^3^, accounting for only a few voxels in an fMRI image. This makes delineating the responses of these two distinct nuclei quite difficult [68,70]. Studies that examined MDD patients compared to matched controls have yielded mixed results [10,71]. Contrary to expectations, MDD patients do not show a noticeable increase in baseline activity or hyperactivity in depressive states. In these neuroimaging studies, BOLD signals were measured during Pavlovian conditioning when the association between symbols that indicate the probability of obtaining monetary incentives or disincentives, and shock, were being made. One of these studies paralleled findings from animal studies that the habenula activity increased in control subjects when a negative event, the chance of receiving a shock, was high. In the MDD patients for the same task, the habenula activity decreased instead, which is difficult to interpret [10]. A second study also reported no significant difference between the activities of habenulae in healthy controls and MDD patients [71]. However, the second study found an overall positive correlation between the positive prediction error-related habenula activity and the frequency of depressive episodes in the participants. 

What these results mean in the broader context of the research in this area needs further analysis. As the authors of these studies point out, caution is required when interpreting human studies of this kind and when comparing the results from animal studies. To draw robust conclusions on the role of the lateral habenula activity in MDD patients, at least four methodological refinements are necessary. First, fMRI resolution for structures as small as the habenula can be faulty even after careful quality control during registration. An exploration of new methods, such as those adapting computational geometry to compare spatiotemporal dissimilarities, may be needed for disambiguation [72]. Better methods are also needed to differentiate the medial and the lateral habenula activity as their activity could be divergent in these tasks and may further complicate interpretation [10]. Larger sample sizes or a meta-analysis of standardized experimental designs are also necessary to increase confidence. Fourth, research groups may also need to evaluate and correct for the possibility of workflow-dependent observational and interpretational bias that has been recently described as being a common issue among functional studies [73]. Apart from these technical considerations, there are other possibilities for the differences between human and animal studies. The BOLD signal recorded in the fMRI study is an abstraction of complex neural activity—of afferents, local inhibition, and spiking efferents [74]. Therefore, an average response is difficult to untangle. The lateral habenula has a complex local circuitry [24], which makes deconvolving, and interpreting the BOLD signal from this region even more difficult. As a large number of neuron types have been reported in the habenula [75], it may also be overly simplistic to interpret an average activity when comparing depressive and non-depressive states. Another reason, as pointed out previously [10], is the possibility that although the metabolic activity in the habenula may be inversely correlated to depressive states, the extent to which this holds may depend on chronic conditions and the state of remission of the subject [32,33], and is therefore dynamic. Finally, the experimental design for human experiments that can reveal the type of bursting activity in a subset of neurons of the lateral habenula observed in MDD rodent models is also not easy to implement and requires further refinement. 

## 5. Perils of Reliance on Single Models to Understand the Neurobiology of Habenula Function in Depression

The use of animal models to dissect the underpinnings of human brain disorders is irreplaceable [76], yet studies have to fulfill a tall order of requirements where differences in cellular physiology, species differences, and the differences due to the complexity of the circuitry need to be taken into account while generating meaningful, transferable knowledge. The studies on MOR and NMDAR described above both used cLH rodents [13,64]. While this model effectively captures many aspects of a depressive state in rodents, whether it adequately represents the cellular and physiological changes associated with MDD in humans in its entirety is debatable. Further, as noted by many researchers, certain natural behaviors such as freezing, escaping, or fighting are altered significantly in animals reared in laboratories for prolonged periods [77,78]. Even if it is argued that relative differences are being examined between two conditions, the generalizability of the findings can be impacted as such alterations are subject to “floor” and “ceiling” effects. Expanding the types of phenotypes examined, and testing many more species broadly, in ecologically naturalistic settings, can overcome the concerns of studying species-specific evolutionary trajectories in singular animal models. In the process, as discussed in a recent review on this topic, it will also deliver more generalizable knowledge and conditions applicable to humans [77]. Comparing and contrasting results across a plurality of animal systems and experimental designs that incorporate behaviors animals normally exhibit in the wild, such as social isolation, social defeat, and dominance hierarchy-dependent stresses will further improve the predictive validity alongside the construct validity of systems used to model depressed states [11]. Some studies have also begun to explore other paradigms, such as symptoms resulting from maternal separation in rodents [60]. More such diverse paradigms are necessary to improve our understanding of the transferability of the accompanying neurobiological changes to the biological changes associated with psychiatric disorders in humans. We acknowledge that starting completely new research programs is neither feasible nor practical. However, adapting already existing models that emulate ecologically naturalistic settings, in tandem with the assays standardized on lab-reared animal models, is both warranted, and plausible. For instance, prairie voles form long-term social bonds with mating companions and display many depressive-like behaviors when separated from their partner [79]. Studying the underlying factors modulating such natural behaviors will be a beneficial complement to current studies using animal systems and will go some distance in addressing the concerns of overinterpretation of animal behavioral data. 

The second point of caution is with respect to the interpretation of results from the standard practice of activating or inactivating neurons to understand how their (in)operation affects specific aspects of a given behavior [80]. Such studies in recent times have been bolstered by the use of optogenetics. They have also been crucial in the study of the habenula function in depression [57]. The ability to control neuronal activity with light is a powerful tool. However, researchers need to be cautious and avoid the trappings of faulty logic [81]. For instance, a claim that “neuron X is sufficient to elicit behavior Y” has the danger of treating the neuron X in isolation, and suggests the highly unlikely possibility that neuron X is the only neuron required in the entire brain for the execution of the behavior [81]. Though this is a universal problem and may appear to be hair-splitting of the implied meaning, it becomes a weak scaffold for future studies reliant on these findings. The same is also applicable to studies employing such manipulation to study depression [14]. Bursting patterns of neural activity could trigger depressive-like behaviors in most mice in this study; however, as the activity of other regions or other neurons is not monitored, the exact physiological conditions of the brain that may result in MDD in an individual are difficult to circumscribe even from such a convincing demonstration of immediate effect. How the dynamics of the homeostatic processes and compensatory changes operate continuously in these neurons and the downstream circuits in response to the increase in the frequency of bursting type pattern of electrical activity is unclear. How that impacts the overall manifestation of depressive states becomes the next question that needs further investigation. More studies, in other animals and humans, and with a variety of experimental designs, will increase the confidence in the interpretation that the pattern of the electrical activity of the lateral habenula neurons is the central player in triggering a depressive state. 

## 6. Comorbidity of Depression with Neuropsychiatric Disorders: An Underexplored Avenue of Research 

While such detailed investigations at the physiological level are essential to devise pharmacological interventions for depression, a complementary approach practiced to a lesser extent is to conceptualize the phenomena and related deficits more broadly. Depressive behaviors have been noted to be often comorbid with a few neuropsychiatric disorders [2,3,5,6]. Smoking in youths in the past year, for example, is highly predictive of ongoing major depressive episodes [2,82]. Moreover, depressed youths are more likely to transition from experimenting to lifetime tobacco use than non-depressed individuals are [83]. Whether the anxiety relief experienced by many when smoking is the primary reason for such associations, or whether there are more fundamental, deeper biological reasons for the co-occurrence, is unclear. In general, psychiatric diagnostics depend on the psychological and somatic symptoms reported by the individual. It is, therefore, possible that the diagnosis of two psychiatric disorders being comorbid could in actuality be a misclassification due to the presence of overlapping cognitive symptoms, artificially inflating the rate of comorbidity. For example, the diagnosis of generalized anxiety disorder (GAD) has been shown to be comorbid with major depressive disorder (MDD), ranging from 40 to 98% of the time [84]. Even though the criteria used for each diagnosis are different, there are a number of overlaps. Overlapping psychopathological criteria include fatigue, sleep disturbances, diminished ability to concentrate, and increased levels of agitation [84]. One way to distinguish if there is indeed shared pathophysiology between conditions is to improve diagnostic criteria with an emphasis on criteria that are unique to a disorder. Irrespective of these necessary improvements in categorization, the co-occurrence of these symptoms itself is indicative of shared psychopathological maladaptations and dysfunction in common molecular or genetic players in these disorders. Therefore, interventional strategies that take into account such possibilities and a broader perspective are highly relevant. 

One commonality between symptoms of different conditions could be the neuromodulator system impacted. A common substrate altered in depression, chronic pain, and insomnia for instance is the dopaminergic function. Its role in arousal has been hypothesized as the potential common factor for the high frequency of the concurrence of these three disorders, each likely exacerbating the symptoms of the other conditions [6]. Anxiety and depression in a similar manner may have the GABAergic system and the GABA_B_ receptors as the common substrate whose function is compromised [85,86]. GABA_B_ receptor antagonists in general show antidepressant effects, while temporal lobe epilepsy patients with depression and anxiety show an increased expression of GABA_B_ receptors as determined by autoradiography studies [86]. The role of this receptor in reward perception, especially that of nicotine reward, is also well studied [86,87], which suggests it to be one point of molecular convergence between addictive disorders, anxiety disorders, and depressive states. Interestingly, hyperactive lateral habenula might also have a role in the manifestation of comorbid disorders. A study investigating the maladaptation of the glial glutamate transporter GLT-1 in changing the activity of lateral habenula neurons in an alcohol-withdrawal rat model found that systemic administration of *ceftriaxone*, an antibiotic known to increase GLT-1 expression, normalized the hyperactivity of lateral habenula neurons in slices, and reversed depression-like and anxiety-like behaviors in rats undergoing alcohol-withdrawal [88]. The lateral habenula could also be the neural circuit link that connects the Alcohol Use Disorders (AUD) comorbidity to depression and anxiety. Research into the circuits where such molecular convergences could play out is anticipated and has been theorized, but is still in its infancy [89]. The physiological conditions that differ between the susceptible and the resistant to comorbid disorders have not yet been fully elaborated. As the study of alcohol withdrawal rats shows [88], new insights and novel roles for molecular players can be discovered when studies directly focus on comorbidity. These may go amiss and remain unanticipated when studies regard each disorder independently. 

Recognition of conditions where mental illnesses are comorbid with depression is also critical to improve the overall clinical outcomes. Preventive measures to counsel subjects who are at high risk for a second disorder when presenting symptoms of the first among comorbid disorders could enhance treatment objectives and subject well-being. Treatment of one of the comorbid diseases may have positive effects on the treatment of other diseases if a common element is targeted. For example, in a study where Cognitive Behavioral Therapy (CBT) was used to alleviate symptoms of social anxiety disorder, a decrease in depressive mood episodes was also reported [5]. Similarly, the reduction in symptoms of depression comorbid with insomnia was observed in treatment with antidepressant medication *escitalopram* in an initial evaluation study of combining pharmacotherapy with CBT [7]. More such studies alongside animal studies where these related phenomena are modeled together rather than as separate ailments are needed for this line of investigation to bear fruit. 

## 7. Psychedelics—A “New” Approach to Antidepressant Treatment

In the context of studying depressive states holistically, psychedelic research has resurfaced in the past few years as an important avenue to seek solutions, after decades of research attempts being stifled by regulatory constraints. The modern era of psychedelic research can be traced back to the well-documented discovery of the properties of the synthetic psychedelic lysergic acid diethylamide (LSD) in 1938 [37]. After an initial period of exploration post the discovery, due to the abuse of the drug and an increase in the societal stigma associated with its use, LSD and several other psychedelics were placed under the controlled Schedule I category of substances, making research with it difficult to conduct in most parts of the world until recently [37]. Despite these impediments, and aided more recently by a relaxation of conditions, a large body of studies have now accumulated, which together suggest that psychedelic use alongside traditional behavioral and cognitive therapies can radically transform the treatment of many mental ailments including addiction to substances, generalized anxiety, clinical depression, and post-traumatic stress disorders [37]. A new generation of behavioral health researchers has begun to re-examine the safety concerns and the efficacy of these substances when administered under medical supervision. Overall, their findings suggest that these substances may provide the long-sought breakthrough in the treatments available for mental illnesses. These developments, along with recent advancements and the promise of psychedelic use in the treatment of brain disorders, have been reviewed in a special issue of Neuropharmacology titled “Psychedelics: New Doors, Altered Perceptions” [90]. 

An important step has been the recognition of the potential of psychedelics in treatments beyond their use as a medication to reduce the despair and suffering of patients in palliative care coping with a terminal illness [91]. For example, a recent study examined the resting-state functional connectivity of the brain after a single, high-dose of psilocybin administered to 19 treatment-resistant depression patients. The team examined symptoms and collected fMRI scans 1-day post-treatment to examine the “after-glow” that subjects report experiencing after the administration of psychedelics, and 5 weeks post-treatment to examine the long-term impacts [38]. While it is impressive that psilocybin treatment showed antidepressant effects at the 1-day post-treatment time point itself (in comparison to SSRIs that can take 4–6 weeks on an average), remarkably, the effects were seen to persist 5-weeks later in most subjects. All but one subject of the 19 showed statistically significant changes 5 weeks post-treatment in the mood on a 16-item Quick Inventory of Depressive Symptoms (QIDS-SR16) survey [38]. The increase in the cerebral blood flow was similar to that observed during LSD and Ayahuasca treatments [92,93]; however, the functional connectivity or the integrity of the default mode network (the DMN) was increased compared to pre-administration scans and can be categorized as being normalized to healthy volunteers. This increase remained at the 5 weeks timepoint for returning subjects. The authors interpret this as a “reset” of the DMN. Wherein, the activity of participating modules disintegrates initially on the administration of the psychedelics, then it is reintegrated, allowing the resumption of “normal function”, unlike their operation in a depressive state [38]. Initial reports from a different cohort of subjects suggest the effects may last for up to 6 months, a highly unusual long-lasting effect when compared to other antidepressants [40]. 

Each psychedelic has a different profile of effects with a fair degree of overlap [39,92,93,94]. The reason for the reset effect described best for psilocybin, could be the increased globular connectivity that has been described as the experience of “expansion of the mind” under the influence of some psychedelics. LSD, for instance, has also been shown to decrease modular connectivity between neural networks while simultaneously increasing global connectivity between high level-cortical regions [95]. This change correlates with the self-reported experience of “ego-dissolution” [95]. Psilocybin, in a similar fashion, increases global communication, which has also been described as the overall increase in the between community (of brain regions) communication rather than within community communication [96]. The “entropic brain” hypothesis encapsulates the phenomenon in the most understandable form, where the entropic metrics are related to both richness-of-content and the diversity of the subject’s experience. These experiences increase under psychedelic use [94]. Thus, greater globular connectivity might mean that the subject is having conscious experiences that engage a larger number of brain regions in a manner not recruited normally [94]. Exactly why this expanded consciousness experience has the desired therapeutic effects is not completely clear. One proposal is that, in some mental illnesses, the networks engage in a particular manner and this becomes rigid and resistant to change. These high-level priors may be “relaxed” under psychedelic use. This decomposition of the existing network connectivity allows for a recalibration of these networks, providing relief in a range of psychiatric ailments where the priors or “beliefs” interfere with normal function [94]. The “reset” mechanism outlined in the study on psilocybin as an antidepressant is consistent with this proposal [38]. In support of such a proposal being a plausible explanation, this study also found that the self-reported intensity of the mystical experience was predictive of the antidepressant effect experienced [38]. 

While the research on potential mechanisms and symptom-level explanations is being investigated actively, a new animal study coincidentally connects these findings in humans to the lateral habenula. A recent study that modeled treatment-resistant depression in rats used optogenetic manipulation to silence the activity of the lateral habenula and examined the rodent DMN using high-resolution rodent fMRI [97]. This perturbation in cLH rats, that the authors categorize as “negative cognitive state” rats, resulted in an overall reduction in the connectivity of the DMN in the rats, similar to the initial effects observed after administration of psychedelics [92,95]. The authors of this study speculate that the anterior cingulate cortex, in the anterior DMN involved in monitoring negative reward, may be receiving indirect input from the lateral habenula via the ventral tegmental area. More importantly, as the authors of the rodent study discuss, recent high-resolution fMRI imaging studies suggest that the lateral habenula is functionally connected to the DMN [98,99]. Therefore, perturbations to its activity can impact the functionality of the DMN (Figure 1). These findings thus connect depression-related studies at two different scales—at the scale of the physiological activity of habenula neurons and at the network-level scale of psychedelic effect. Understanding of a circuit-level explanation of the immediate and long-term effects of psychedelics is critical and insights into it are just beginning to emerge [37,90,94]. Another topic of great interest for its application for treatments at present is also to understand the molecular mechanisms by which psychedelics bring about their effect. 

## 8. Molecular Mechanisms of Psychedelic Action and the Future of Antidepressant Discovery

Pharmacologically, both Psilocybin and LSD, and other molecules that act via the serotonergic system have been categorized as psychedelics [94,100,101,102]. Their function, primarily as agonists of serotonin receptors (5-HTR) has been confirmed in a remarkable study that used multi-modal imaging to simulate serotonin receptor activity in the whole brain, generating the closest match to the brain on LSD to date [102]. They achieved this by overlaying Positron Emission Tomography (PET) imaging-derived maps of serotonin receptor density distribution on fMRI images of healthy volunteers on LSD. Several other studies also point to serotonin receptors, particularly to 5-HT_2a_, to be the molecular target of psychedelics [94]. As pointed out by the author Carhart-Harris, it is indeed an extraordinary finding that a single molecular target could underlie the profound and varied subjective experience of psychedelics. What happens downstream to the receptor activation is not fully understood yet. Given that a single treatment with psilocybin can have sustained antidepressant effects even 5-weeks after the treatment [38], it has been hypothesized that long term changes could include transcriptional changes and activation of specific immediate early genes (IEGs) in the receiving neurons [42]. As IEGS are expressed within minutes after an intense or unusual neural activity, they are not only used to identify active neurons but are also the starting point to decipher the cascade of events after exposure to a substance. Brain-derived neurotrophic factor (BDNF) and activity-regulated cytoskeletal protein (arc) have been associated with psychedelic exposure; however, a large gap still exists in the characterization of the downstream signaling pathways involved as the circuitry activated is complex [42]. 

One way forward for this research is to use the “mini-brains” or cortical organoids that show complex cellular organization and physiological responses [103]. Cerebral organoids have an exceptional potential to help bridge the gap between animal studies and human trials. Organoids are 3-D cellular structures that show realistic micro-anatomy and that are capable of self-renewal and self-organization. Recent studies show that cerebral organoids derived from human pluripotent cells progress through similar developmental trajectories and can develop to features resembling a 10-week old human brain [104,105]. Organoids allow for a more controlled environment to perform investigations of neuropsychiatric illnesses at a molecular and cellular level compared to animal models or ex-vivo primary cultures. Importantly, as they are directly derived from human pluripotent cells, it circumvents questions raised about species-specific differences in the brain tissue. Organoids have become a useful platform to study cellular interactions between different cell types that are not as easy to conduct in-vivo; for example, for studies on glial–neuron interactions important in the pathology of neuropsychiatric diseases [12,106]. Findings such as the development of microglia important for the synapse formation innately within cerebral organoids support the possibility of using cerebral organoids for such research [107]. Moreover, cerebral organoids maturing over a period of months show many complex features with dendritic spines and spontaneously active neuronal networks, thus allowing for the study of a “circuit” level analysis of the effects of antidepressants [108]. 

The use of the cerebral organoids to uncover the molecular mechanisms of psychedelic action overcomes many limitations imposed by the lack of appropriate animal models and regulatory restrictions [109,110,111]. Detailed studies with single-cell level “omics” analyses can be performed to characterize the transcriptomic and proteomic profile after different periods of exposure to a psychedelic. A recent study demonstrated the capability of such an approach by examining the effects of 5-methoxy-N, N-dimethyltryptamine (5-MeO-DMT) on the proteome of cells in human cerebral organoids. Dimethyltryptamines (DMTs) are naturally occurring psychoactive agents present in brews and Ayahuasca used in Shamanic religious practices for their antidepressant properties [112]. Using mass spectrometry to perform unbiased proteome analysis, the study found that a single exposure for 24 h modulated several signaling pathways. Signaling cascades indicative of increased anti-inflammatory effects and modulation of proteins implicated in spine morphogenesis and function, such as those participating in protrusion, microtubule dynamics, cytoskeletal reorganization, and Long Term Potentiation (LTP), were detected [111]. In addition, the study also found inhibition of pathways associated with cell death, thus providing some very new insights into the potential molecular mechanism by which DMT likely produces its effects. Another direction for the use of this approach is to simulate the effect of psychoactive agents and stimulants in utero or on the developing fetus [106]. Such molecular characterization of drug responses to psychedelics could yield equally exciting insights into their effect on different sub-populations of neurons and their effects on synaptic plasticity and neural proliferation.

Another promising direction anticipated in the antidepressant research is to derive organoids for use in precision medicine type studies and examine the potential effects of antidepressants in the context of individual risk variants and known genetic susceptibilities [113]. They may also be particularly well suited to study the genetics of a combination of alleles that each have a small effect on the phenotype, and are difficult to recreate in animal models. For instance, it has been shown that only approximately 50% of patients respond to SSRIs the first time, and less than one-third go into remission [114]. Failure to respond to the initial course of treatment can mean an increased risk of suicide, distress, and loss of productivity among the effected [115]. Predictive biomarkers that can help stratify potential responders can dramatically reduce such risk [116]. However, the clinical applications of such biomarker-based selection can be hampered due to small samples and effect sizes [117]. Patient-derived cerebral organoids offer a potential way to compliment the knowledge of current biomarkers with tests to examine the potential efficacy of the treatment. Assays that screen for multiple drugs and their combinations could improve throughput and arrive at the appropriate course of action faster [113,118]. For such applications to work, however, issues such as the time taken to grow mature cerebral organoids (4 to 6 weeks) and the cost-effectiveness of the strategy have to be taken into account. Despite these considerations, the organoids technology provides a new dimension in psychedelic research and an exciting prospect for the discovery of therapeutic aids for MDD and treatment-resistant depressive states.

## 9. Conclusions

Lateral habenula is one of the most intensely researched brain regions currently to find remedies for treating depression and several comorbid neuropsychiatric disorders. Methodological refinements in human studies, alongside an expansion of the animal systems used to model depression, are necessary to understand the role of this brain region in these disorders. Studies that address the sparsity of models investigating the common underlying molecular factors in comorbid disorders have the potential for discovering new therapeutic approaches. At the same time, understanding the antidepressant mechanism of psilocybin and psychedelics warrants scientific priority as the use of these agents has the possibility of providing profound insights not only into how to “rewire” a brain in depression but also into the normal operation of brains. We think it will also reveal the exact nature of the connections between these two areas of research that appear to be emerging (Figure 1). 

## Figures and Tables

**Figure 1 ijms-21-06525-f001:**
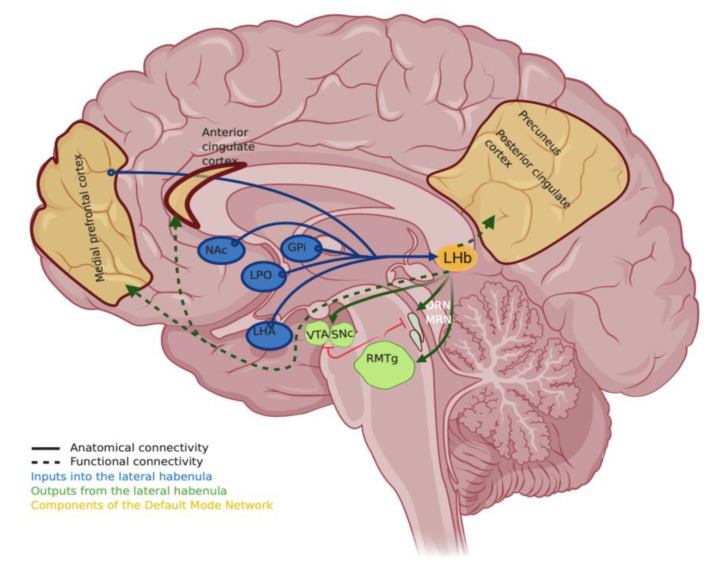
The cartoon shows a few major inputs (in blue) and outputs (in green) of the lateral habenula based on [11]—NAc, Nucleus Acumbmens; GPi, globus pallidus internus; LHA, lateral hypothalamic area; LPO, lateral preoptic area; MRN and DRN, medial raphe nucleus or dorsal raphe nucleus; RMTg, rostromedial tegmental nucleus; SNc, substantia nigra pars compacta; VTA, ventral tegmental area. Components of Default Mode Network (DMN, in golden) whose connectivity and gene expression profile changes by psychedelics include precuneus/posterior cingulate cortex (PCC), medial prefrontal cortex (mPFC) and the anterior cingulate cortex. The DMN cartoon is based on [42,93]. Solid lines show major anatomical connectivity described from animal and human studies, while dashed lines show functional connectivity based on recent human studies [97,98,99]. Created with BioRender.com.

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
