# Peer review of "Total Recall: Lateral Habenula and Psychedelics in the Study of Depression and Comorbid Brain Disorders"

_ijms, 2020, doi:10.3390/ijms21186525_

Round 1

Reviewer 1 Report

In this review titled "Total Recall: Lateral habenula and psychedelics in the study of depression and comorbid brain disorders”, Vitkauskas and Mathuru propose an original perspective aiming to bridge the central role of the lateral habenula (LHb) in the pathophysiology of depressive disorders with the recent interest on the anti-depressant action of 5HT Rs-acting psychedelic derivate (i.e. LSD, psilocybin). The authors also try to review the potential role of the LHb in disorders comorbid with depression. Finally, they highlight the advances in the use of cerebral organoids, indicating the latter as a preferential road to follow to increase translational potential.  Overall, the review is pleasant to read and give an immediate sense of the author’s point of view. I personally like the flowing of the arguments from the description of the LHb role in mood disorders (reporting both animal studies and clinical evidence) to the intriguing possibility, here raised, that antidepressant effect of psychedelics can be partially explained by an action on LHb activity. I was instead confused by the paragraph on the comorbidity. I do not immediately see what it adds to the flowing of the present review. I also believe that such topic may need a more careful in-depth analysis.

Here some specific comments/suggestions:

1) Regarding the description of the lateral habenula role in depression (animal models studies) the authors should consider to acknowledge work other than the one provided by the Hailan Hu lab and Roberto Malinow lab (i.e Cerniauskas et al 2019 Neuron; Huang et al 2019 Neuron; Lecca et al 2016 Nat Medicine; Tchenio et al 2017 Cell comm; Seo et al 2018 Mol Psy).

 2) “…in a behaving monkey have helped contextualize older observations of habenula  inhibitory efferents to substantia nigra and ventral tegmental area dopamine neurons [50,51], as well as  to the serotonergic dorsal raphe nucleus [15].” Not correct. Neurons in the LHb are virtually exclusively glutamatergic. The inhibitory power of LHb over downstream targets is mainly occurring via indirect modulation (since a greater direct LHb input impinge on GABA interneuron in VTA as well as raphe or GABA cells in the RMTg that in turn controls Da and 5Ht cells).

3) “In support of the hypothesis that high-frequency DBS likely works by reducing the hyperactivity, one study showed that low-frequency stimulation conversely activates lateral habenula and produces an increase in depressive-like behaviors in rats [35]”. Here the authors should also cite Tchenio et al 2017 (Nat Comm), a work where the authors directly test the effect of high frequency DBS on LHb neuronal activity.

Reviewer 2 Report

Overall, we found this review to be very interesting and informative. The topics covered, including lateral habenula (LHb) perturbation in depression and the use of psychedelics as therapeutics for psychiatric disorders, are relatively new and exciting areas of research. However, we are not sure how the title “Total Recall” relates to the article as it does not discuss memory processes, therefore we would suggest an alternate title. There are a few grammatical or typographical errors, and several references are not complete citations, such as 78, 81, 88, among others. On line 182, the authors reference source 69, which does not seem to fit with the text they attribute to it. A particularly important error is on lines 66-67, which states that “the lateral habenula neurons fire when expected punishments are omitted.” Instead, LHb neurons fire when expected rewards are omitted. Such errors contribute to a sense of sloppiness or hurriedness in the writing.

In the section “Targeting the hyperactivity of lateral habenula provides antidepressant effects in animals,” we think the authors may be overinterpreting animal behavioral data, particularly describing exploration in an open field as motivational behavior when this test is primarily used to measure anxiety-like behavior. Similarly, immobility in the forced swim test is primarily used to measure antidepressant-like effects in rodents, and many do not believe it is a depressive-like behavior (Molendijk & Kloet, 2019). Finally, on line 125, the authors use the term “MDD mice” to describe stressed mice eliciting depressive-like behaviors. Additionally, the authors discuss the importance in moving towards studying animals in more “ecologically naturalistic settings,” and while this is an important consideration and noble goal, they may want to comment on the feasibility of doing so. It would require starting over from scratch as the vast majority of data in the literature that would inform future studies were carried out in animals housed under standard laboratory conditions. Therefore, it would require a herculean effort to replicate those data in animals housed in naturalistic settings. Furthermore, it is unclear how animals in a natural environment might respond to unnatural stressors.

When discussing disorders that are often comorbid with major depressive disorder (MDD), such as anxiety, substance use disorder, and post-traumatic stress disorder, the authors primarily focus on potential shared neural mechanisms between these disorders. It is likely that there are shared neural mechanisms but given that psychiatric diagnoses have no biological basis, but are syndromes based on symptom clusters that occur together, overlapping symptoms likely contribute to comorbidity and shared neural mechanisms. Therefore, we think there would be significant value in discussing shared symptoms, which would certainly inform potentially-shared neural mechanisms. Additionally, this would also inform the upcoming section on psychedelics, and how they may be effective treatments for multiple disorders, not just MDD, due to such overlap in symptoms and mechanisms. In particular, the overlap of cognitive symptoms, such as maladaptive thought processes, would be particularly beneficial to discuss since it would tie into the “reset” of the default mode network (DMN) by psychedelics and subsequent reshaping of thought processes through therapy.

While the authors give a thorough explanation of the DMN theory of psychedelic therapy, it would be beneficial to expand on this when discussing mechanisms of psychedelics rather focusing so much on molecular mechanisms. No doubt understanding the molecular mechanisms of psychedelics in the context of their therapeutic effects is important, but circuit mechanisms would be equally or even more relevant to discuss in this review. Our main criticism of the review is the lack of connection between the reviews on LHb dysfunction in MDD and the therapeutic effects of psychedelics. Focusing more on circuit mechanisms rather than molecular mechanisms would provide an opportunity to tie these two areas of research together and hypothesize about how the LHb could be affected by the presumed reset of the DMN or if activity changes in the LHb play a role in the reset, as alluded to by the mention of the study done by Hohenberg et al. (2018). Finally, the addition of at least one figure depicting the relevant circuitry of the LHb and the behaviors those circuits are believed to contribute to or modulate would be highly beneficial. This would also be a good place to illustrate the relevant DMN circuitry and potential connections between the LHb and DMN.

Hohenberg, C. C. von, Weber-Fahr, W., Lebhardt, P., Ravi, N., Braun, U., Gass, N., Becker, R., Sack, M., Linan, A. C., Gerchen, M. F., Reinwald, J. R., Oettl, L.-L., Meyer-Lindenberg, A., Vollmayr, B., Kelsch, W., & Sartorius, A. (2018). Lateral habenula perturbation reduces default-mode network connectivity in a rat model of depression. Translational Psychiatry, 8(1), 68. https://doi.org/10.1038/s41398-018-0121-y

Molendijk, M. L., & Kloet, E. R. de. (2019). Coping with the forced swim stressor: current state-of-the-art. Behavioural Brain Research, 364, 1–10. https://doi.org/10.1016/j.bbr.2019.02.005

Reviewer 3 Report

This review discussed the role of Lateral habenula and psychedelics in the depression, and gave out a general idea of how lateral habenula and psychedelics involved in depression treatment. I only have few minor concerns.

1, Hyperactivity of LH is a major change happened in MDD. There were intrinsic and pre- /post-synaptic changes in LHb, which also contributed to depression. The authors may discuss. 

2, Except DBS and ketamine, there were more reports or therapeutic angles about targeting LHb for antidepressant effects. The authors should have more detailed discussion.   

3, The authors mentioned both targeting LHb hyperactivity and psychedelics have antidepressant effects, and put them in the same manuscript. Was there any connection between these two? Can psychedelics administration cause LHb activity change?

Round 2

Reviewer 2 Report

The authors have adequately and comprehensively addressed our concerns.